# Prospective Study of Urinary Stone Formation in Pediatric Patients with Acquired Brain Injury: A Focus on Incidence and Analysis of Risk Factors

**DOI:** 10.3390/nu17050883

**Published:** 2025-02-28

**Authors:** Sara Galbiati, Federica Locatelli, Fabio Alexander Storm, Marco Pozzi, Sandra Strazzer

**Affiliations:** Scientific Institute IRCCS Eugenio Medea, 23842 Bosisio Parini, LC, Italy; sara.galbiati@lanostrafamiglia.it (S.G.); federica.locatelli@lanostrafamiglia.it (F.L.); fabio.storm@lanostrafamiglia.it (F.A.S.); sandra.strazzer@lanostrafamiglia.it (S.S.)

**Keywords:** urinary stones, urinary tract infections, enteral nutrition, brain injury, rehabilitation

## Abstract

**Background/Objectives**: Little is known about the factors linked with nutrition, infections, and physical activity, which may influence urinary stone formation in patients with acquired brain injury. Previous studies have demonstrated that enteral nutrition mixtures rich in sodium and poor in calcium may promote stone formation in pediatric patients, but a confirmation study is lacking. Moreover, the occurrence of urinary stones and heterotopic ossifications has not been studied regarding incidence. We thus conducted a prospective observational study in an unselected pediatric population with acquired brain injury, to estimate the incidence of urinary stones and heterotopic ossifications and analyze the associated factors. **Methods**: Prospective observational study: We recruited all patients with enteral nutrition consecutively admitted to our brain injury rehabilitation unit during a 5-year time-frame. We collected clinical data regarding nutrition, infections, blood and urine exams performed, neurological examinations, and physical examinations. **Results**: The prospective design allowed us to observe that no patient developed heterotopic ossifications, while urinary stones were found in 12.5% of patients and gravel in 14.6%. Factors associated with stone formation were having a worse subacute GCS, having done intense physical activity before injury, receiving bladder catheterizations, having a higher urine pH, and having higher blood potassium levels. The composition of the enteral nutrition did not influence stone formation, although the nutrition mixtures contained levels of vitamin C and proteins considerably higher than the recommended reference ranges. **Conclusions**: We have provided an observation of the incidence of urolithiasis in pediatric patients in rehabilitation, which was lacking from the literature. Enteral nutrition, at the amounts received by the patients studied herein, does not seem to have a role in stone formation. We identified a set of risk factors that can be useful for clinicians to pinpoint patients at an increased risk of developing stones.

## 1. Introduction

The literature on the issue of urolithiasis in children focuses on children with metabolic disorders of genetic or syndromic origin, as well as congenital neurological conditions [1,2,3,4,5]. Various risk factors include infections [5,6,7], low urinary pH [5,8], hypercalciuria [5,7,9] and/or hypercalcemia [5,7], immobilization [6,7,8,9], obesity [10], bladder drainage [6,8], the use of enteral nutrition [6,8,9], and anticonvulsant medications. Among these, Topiramate is the most frequently mentioned, as it acts as a carbonic anhydrase inhibitor and can lead to hypercalciuria, hypocitraturia, and urine alkalization [6,8,9,11,12].

Regarding stone composition, calcium oxalate stones appear to predominate [5,6,8], with their prevalence varying across multiple studies. Calcium phosphate stones are also frequently reported [8]; the less common struvite stones are observed in a small proportion of cases [6,8]. These studies mainly address congenital conditions where physical and metabolic alterations develop progressively over time.

In cases of urolithiasis secondary to acquired brain injury (ABI), the pathogenesis is different, since a sudden disruption of normal physiology and metabolism occurs. This abrupt change often includes the cessation of daily physical activity and regular nutrition. We found little relevant literature on urolithiasis after ABI, with only a few studies involving patients with spinal cord and/or brain injuries [13,14,15]. In adults with ABI, there are many reports of heterotopic ossification, indicating that the metabolic modification leads predominantly to abnormal calcium deposition [16,17,18]; reports of heterotopic ossification in children are absent, indicating a predominance of calcium loss through urine and consequent urolithiasis.

In a previous retrospective study on urolithiasis in children after ABI [19], we assessed issues related to urinary stones within a retrospective case-control study. We observed only patients receiving enteral nutrition exhibited urolithiasis, while not having a study design apt to calculate epidemiological indices. Moreover, in univariate analysis, these patients were older, had higher body weight, received larger volumes of hydration and nutrition, had worse Glasgow Coma Scale (GCS) scores, experienced more urinary tract infections (UTIs), and exclusively underwent catheterization [19].

The importance of urolithiasis in patients with ABI is high, since urinary stones can cause severe and recurrent pain and spasms, symptoms that are challenging to interpret in patients with severe cognitive or communication impairments [3]. Pain can be managed with drugs that in several cases are opioids, which may compromise consciousness recovery and pose other health threats [20]. Spasms may trigger or exacerbate sympathetic paroxysmal episodes [21], complicating the clinical course and delaying or hindering access to rehabilitation. Moreover, there is a strong reciprocal correlation between urolithiasis and urinary tract infections [19], an issue that demands antibiotic use to target microorganisms that are often multidrug-resistant.

For these reasons, it is important to prevent, monitor, and adequately treat urolithiasis in pediatric patients who suffered ABI, in order to prevent complications and delays in rehabilitation.

To expand the knowledge base on this issue, we report on the results of a prospective observational study on the incidence of urolithiasis in post-acute pediatric patients with ABI and on the correlated clinical factors.

## 2. Materials and Methods

This work is comprised of two elements. The first is a prospective observational study, the second is a pooled post-hoc analysis, formed with patients enrolled both in the prospective observational study and in a previous retrospective case-control study [19].

The prospective observational study was conducted during the five years of 2019–2023. Caregivers of all patients admitted for rehabilitation to the acquired brain injury unit of IRCCS Eugenio Medea (LC), Italy, were asked for willingness to participate by providing an informed consent. Enrolled patients were treated following the routine clinical practice; no experimental procedure or supplemental diagnostic exam was conducted. All study procedures were carried out following the principles set out in the Declaration of Helsinki. This study received formal approval from the institutional ethics committee on 23 March 2018 (protocol 04/18-Oss).

Patients were evaluated for inclusion in the prospective study based on consecutive admission to the ABI rehabilitation unit, without a priori selection.

Inclusion criteria were being admitted to our brain injury rehabilitation unit during the years 2019–2023, being in a sub-acute phase (more than one, less than three months from brain injury), and receiving enteral nutrition either predominantly or exclusively.

Exclusion criteria were having a previous history of urinary stones, having a known genetic predisposition to form urinary stones, and being already under prophylaxis or treatment for urolithiasis.

Clinical data required for this study were collected prospectively and curated and analyzed after the end of the study. They included data regarding ABI and neurology, medical devices use, drug therapies, and, in particular, antiseizure drugs, urinary physiology, and physical medicine assessments.

Clinical data on the baseline situation were collected at admission to rehabilitation; data on the follow-up picture were collected at the moment of diagnosis of urolithiasis. Analyses carried out are mainly descriptive, including the incidence rate of urinary gravel and stones, the distribution of clinical variables, and the study of correlations among clinical variables. We used T-test, χ^2^ test, and Pearson and Kendall’s correlations. As several variables were inter-correlated, we simplified the dataset by factor analysis, with parameters Eigenvalue 1 and varimax rotation, and required sampling adequacy > 0.7 for the whole model and retained variables.

The second element of this work is the recollection, harmonization, and reanalysis of data from a historical case-control cohort on which we published in past years. The purpose of inclusion of older data in the current dataset was to provide more power for analyses. Data across the two datasets were initially checked for consistency, which was passed, demonstrating that the two patient groups were not inhomogeneous. Then, the datasets were merged, and mismatches were amended.

We finally checked for the possible contribution of clinical variables in determining the occurrence of urolithiasis by using a stepwise (*p*-in < 0.05, *p*-out > 0.10) series of logistic regression models with the occurrence of urolithiasis as a dependent variable. In the results, we report the model fit, adjusted pseudo R^2^ and lack-of-fit of the final model, and the list of significant predictors with their respective odds ratios (OR) with 95% confidence intervals. For all tests, significance levels were set at *p*-values of <0.05 and two-tailed. Analyses were conducted using SPSS v.29 (IBM, Chicago, MI, USA).

## 3. Results

### 3.1. Population Characteristics

#### 3.1.1. Clinical Characteristics

In the prospective study, we could enroll 48 patients undergoing rehabilitation for ABI in the sub-acute phase and receiving enteral nutrition. Their age was on average 9.5 ± 1.0 years and 66.7% were males. Patients’ body weight was 29.8 ± 2.7 kg on average.

The origin of ABI was mostly traumatic (29.2%), followed by anoxic/ischemic injury (22.9%), hemorrhagic stroke (14.6%), and encephalopathy of several kinds (25.1% altogether). Injuries were mostly severe, and the acute GCS was 5.9 ± 3.5 on average. At admittance to rehabilitation, the subacute GCS was 9.8 ± 0.4 on average. Regarding devices, 52.1% of patients had a tracheostomy tube at admission to rehabilitation; 4.2% of patients were also capable of oral nutrition, while exclusive enteral nutrition was assumed by nasogastric tube in 27.1% and by gastrostomy or jejunostomy in 68.8%. Regarding physical activity before injury, 39.6% of patients practiced heavy or competitive sports, while after injury 97.9% were bedridden. Besides this, cholelithiasis was present in 12.5% of patients. Regarding urinary tract health, the mean urine pH was 7.1 ± 0.1, 20.8% of patients had the need for catheterism, and 37.5% of patients had urinary tract infections; the prevalent microorganisms were *E. faecalis*, *E. coli*, *K. pneumoniae*, and *P. aeruginosa* (Table 1). With respect to treatment, only two patients were taking antibiotics at admittance, and they did not develop stones.

#### 3.1.2. Enteral Nutrition Characteristics

Patients used a variety of enteral nutrition mixtures, complemented by nutritional supplements. We extracted the nutrition facts of each mixture, added the supplements, and compared the overall nutrition amounts to reference levels. We chose reference levels for nutrition facts following EFSA guidelines, as summarized in Table 2.

Patients receiving daily nutrition amounts above the recommended values were 12.5% for magnesium, 18.8% for sodium, 22.9% for calcium, 25% for potassium, 60.4% for vitamin C, and 70.8% for proteins. Patients received 0.7 ± 0.4 L of water per day.

Given that nutrition mixtures have fixed combinations of nutrients, we chose to statistically simplify the nutrition data in order to use it as a variable in regression models. By performing factor analysis, with an overall sampling adequacy of 0.79 and a Bartlett’s test of *p* < 0.001, we identified a common factor resuming the content of potassium (sampling adequacy 0.90), magnesium (0.83), proteins (0.82), vitamin C (0.73), and calcium (0.72), but not of sodium. For further use in the regression models of this work, we named this factor “non-Na nutrition”, since sodium was statistically excluded from the computation of this factor.

### 3.2. Incidence of Urolithiasis and Associated Clinical Factors

#### 3.2.1. Incidence of Urolithiasis and Associations with Single Clinical Factors

During the prospective follow-up, urinary stones were found in 6 (12.5%) patients, gravel in 7 (14.6%), and considered together as 13 (27.1%) events of urolithiasis (the clinical outcome). In all cases, they were calcium and phosphate based, no uric acid stones were found. Among the 6 patients with analyzable expelled stones, 3 had calcium oxalate stones (of which, 1 with *E. coli* infection), 1 struvite (with *E. faecalis* infection), 2 struvite and oxalate (one with *E. faecalis*, one with *K. pneumoniae* infection). In agreement with the literature, we found no occurrence of heterotopic ossification in children. Table 3 presents a comparison between the groups of patients with and without urolithiasis for clinical variables.

In brief, by descriptive analyses, patients with urolithiasis appeared to be older, more frequently with a tracheostomy.

No significant difference was found between groups regarding the contents of nutrition, nor the number of patients that received nutrition mixtures exceeding the reference values of any nutrition component cited in Table 1. Patients with urolithiasis drank more water, were older, and therefore bigger. Blood and urine exams were performed to check for the presence of biomarkers of urolithiasis; at a descriptive analysis, no association with urolithiasis appeared to be significant, except for a higher urine pH.

We explored the possibility of simplifying patient characteristics that were inter-correlated using factor analysis. Correlations indicated two suitable sets of variables, one being age–weight–water intake and the other being acute GCS–tracheostomy–oral feeding, but neither set met statistical requirements to retain the results of factor analysis.

#### 3.2.2. Associations of Urolithiasis with All Clinical Factors

Moving beyond univariate associations of individual factors with urolithiasis, we attempted to obtain a statistical model predictive of the risk to develop urolithiasis, given the prospective nature of data. We performed logistic regressions on the prospective study dataset, to find variables predicting the occurrence of urolithiasis. Since the best resulting model had a pseudo-R^2^ = 0.14, implying that only 14% of variability was explained, we sought possible alternatives for data analysis. The choice was made to integrate the dataset from our previous retrospective case-control study, in order to boost statistical power at the expense of causality inferences. We initially confirmed that the distribution of clinical variables was not different between the two datasets.

On the aggregated dataset with n = 88, composed of 40 historical case-controls plus 48 prospective patients, we performed a set of stepwise logistic regressions in order to assess the relationship of collected variables with the occurrence of urolithiasis.

Preliminary models were run again to remove inter-correlated variables, which were the same also in the merged dataset. Inter-correlated variables were inserted in turn in the models, maintaining the best one of the inter-correlated sets and discarding the others. From the set age–weight–water intake, age was kept. From the set acute GCS–tracheostomy–oral feeding, acute GCS was kept, and the factor analysis of nutritional facts was confirmed, leaving nutrition simplified into “Na” and “non-Na.

The variables tested in the final logistic regression model were age, acute GCS, subacute GCS, catheterism, gallstones, intense physical activity before ABI, bedridden after ABI, non-Na nutrition, Na-nutrition, all blood exam parameters, urine pH, and urinary tract infections. The resulting final step-model had a pseudo-R^2^ = 0.47 and a global *p*-value < 0.001; the lack-of-fit test was passed. The model retained the following variables with respective odds ratios (OR) and *p*-values, reported in Table 4.

This model demonstrated a possible association between urolithiasis and several clinical, nutritional, and blood and urine exam variables. In particular, the subacute GCS score appeared as a protective factor, which can mean that patients with injuries that are better recovering are less prone to complications; intense physical activity before injury and the need of cathetherisms appeared as strong risk factors, since they imply respectively a greater reservoir of calcium to be lost from bones, and a difficult urinary physiology. Urine pH also resulted as a moderate risk factor, implying that more basic urine is associated with more frequent urolithiasis. Finally, higher K blood levels were associated with more frequent urolithiasis.

## 4. Discussion

Urolithiasis is an internal medicine complication that can arise subacutely after brain injury and may be assigned marginal importance as compared to neurological damage. However, during the rehabilitation process, the onset of stones, or of any problems that cause pain, leads to complex physical responses. These involve, for instance, an increase in muscle tone in patients with spasticity [28,29], or a precipitation of spasms and agitation in patients with compromised consciousness [30], and these motor impairments can influence the diagnosis of low levels of consciousness [31,32]. Moreover, pain can trigger and worsen paroxysmal sympathetic hyperactivity, which is a prominent cause of death and additional disability after brain injury [33,34]. Infections and fever connected with stones also hinder the rehabilitation process, interrupt it, and make the management of polypharmacy regimens even more complicated [35,36]. For these reasons, risks due to the occurrence of urolithiasis must be monitored, especially in the most fragile patients who cannot communicate. Knowing the problem, understanding what facilitates the onset of urolithiasis and advancing prevention and treatment, can ultimately reduce the burden of physical complications and pain, easing the rehabilitation course.

The literature presents almost no evidence regarding the occurrence of urolithiasis in pediatric patients with brain injury. After the publication of our retrospective case-control analysis in 2019 [19], no other article dealing with this topic was published. The dearth of evidence is striking, in view of the relevance that urinary stones can have in the pathogenesis of chronic spasticity and pain [37].

In the present work, we refined our previous research by performing a prospective cohort study on pediatric patients with brain injury, in order to assess the incidence of complications related to urolithiasis and to verify previous retrospective results with a stronger causality relationship. Moreover, studies specific for patients with brain injury are required since these patients experience a sudden change in physical, hormonal and metabolic aspects [38], that make urolithiasis in this condition unique and demand deeper investigation.

To address the issue, we recruited and followed prospectively a cohort of patients who received enteral nutrition, either exclusively or predominantly, since we previously observed that enteral nutrition is strongly correlated with urolithiasis [19]. From the clinical exams and records of these patients, we obtained a baseline observation at admittance to rehabilitation and a follow-up observation when they developed urolithiasis. With respect to our previous work, we aimed to verify causal relationships, an opportunity that was denied by the need to assemble a larger dataset. The resulting mixed retrospective-prospective dataset is still valuable; it allowed us to verify previous results and to also investigate, for instance, the role of bladder catheterization, which had not been assessed before. However, we could not claim any causality effects due to the inclusion of retrospective data.

Notable changes in the prospective study results, compared with the previous historical cohort, regarded nutrition mixtures and dietary supplements. After evaluating the results of the previous publication, the clinical practice was altered at our institution, such that nutrition mixtures with lower sodium content were chosen as standards; it is not surprising, therefore, that sodium had no significant association with urolithiasis in the full dataset analysis. Alongside this change, the reduction of variability in nutrition mixtures resulted in the possibility of simplifying the nutritional content via factor analysis, a statistical procedure which was not possible on the older dataset only.

The clinical implications of this absence of correlation are interesting, because by setting nutrition facts around the values we described, enteral nutrition should be expected to pose a lesser risk for urolithiasis. This hypothesis should be further verified by studying patients not receiving enteral nutrition, who still incur urolithiasis, to be taken as comparison subjects.

The larger size of the present dataset allowed us to observe significant associations between urolithiasis and subacute GCS, with both urine pH and the use of catheterization, and of physical activity before injury, factors that previously resulted in only univariate statistics. The role of these associated factors is meaningful and of clinical utility.

Subacute GCS is a direct indicator of the neurological recovery. Indeed, we observed that patients with a better recovery even before rehabilitation are less prone to complications such as urolithiasis; this may be due to a better health status, or to a consequent minor impact of injury on physiological functions, including the urinary tract health.

The use of catheterization, although not associated with GCS, can be seen as a similar factor, since patients with worse neurological recovery are more prone to require catheterization to favor bladder discharge [39], and catheterization is a semi-invasive technique that increases both the risk of infection and traumatizes the urinary tract, further delaying the recovery of physiological activities [39,40]. Since in the previous retrospective study we found a significant effect from urinary tract infections on increasing the occurrence of stones, it is important to highlight that in our current clinical practice, at variance with the past one, potential cases of urinary tract infection are managed together with urologist consultancy.

In the absence of symptoms, we follow the principle of avoiding unnecessary antibiotic treatment. This approach has the advantage of reducing the spread of antibiotic resistance and reducing adverse effects.

Alternatives to systemic antibiotic therapy that we often consider are intravesical instillation of antibiotic, improved hydration, and change of urine pH to make the microenvironment unfavorable for bacteria.

Antibiotic treatment is started only in the presence of fever, blood markers of inflammation, positive abdominal echography, urine analysis with bacterial load above 100,000 cfu/mL, and provided the antibiogram suggests viable strategies. The only exception is for emergency situations when rapid intervention with a broad-spectrum antibiotic is advised.

Given this actualized clinical approach, we found no more associations of urinary tract infections with the development of urolithiasis in our study sample.

A factor that we found to significantly increase the risk was a higher urine pH. It is connected with easier stone formation in the general population [41] due to increased stability of calcium phosphate complexes at a more basic pH [42]; we could also confirm its role in the specific setting of pediatric brain injury.

As previously hypothesized, we could now demonstrate a risk associated with intense physical activity before brain injury, which is due to the probable heavier calcium content of bones in athletic persons, that becomes progressively lost after injury and immobility [43], acting as substrate for either heterotopic ossification in adults and for urolithiasis in children.

Of interest, in the larger dataset we found a significant association of urolithiasis with higher blood potassium levels, which might be explained in view of the association with higher urine pH; however, in our dataset, there was no significant inter-correlation between urine pH and blood potassium.

Several limitations apply to the present study. Although this study incorporated retrospective data, the prospective observational component lacked a control group that did not receive enteral nutrition. As a result, it is difficult to directly compare the risk of urolithiasis associated with enteral nutrition. Including a matched healthy control group or patients on different nutritional regimens would provide a clearer assessment of the impact of specific nutrients on stone formation. In our clinical practice, such comparisons were not available since, following our previous study [19], we observed that patients not receiving enteral nutrition did not develop urolithiasis during rehabilitation, and also since we reduced significantly the variability among nutrition mixtures used in our clinical routine.

A method to overcome this limitation would be to analyze the excretion of electrolytes, together with the intake, as well as performing a water balance. These investigations go beyond the observational nature of the present study and would have required a dedicated clinical trial, which was out of our scope.

Another limitation is the failure to provide a causal inference on the prospective study results, for instance, regarding the relationship between urolithiasis and urinary pH, it cannot be assessed whether urinary pH may cause stone formation in our patients.

With the sub-optimal power of statistical analyses, our combined dataset could reach a 47% of explained variability regarding urolithiasis, which indicates either that important factors influencing urolithiasis are still missing from being analyzed or that even larger population sizes are required to provide more sound statistical results; future studies should include a broader set of potential associated factors, including citrate, oxalate, and electrolyte excretion, fluid balance, and others, and aim to better understand the role of urine pH in stone formation.

It is important to stress that in our previous study we observed only patients with enteral nutrition developing urolithiasis. No patient without enteral nutrition developed urolithiasis. This complete correlation indicates that either enteral nutrition is in itself a critical risk factor for urolithiasis, or that the time-frame of recovery during which enteral nutrition can be used, i.e., the sub-acute phase, is the critical one for the development of urolithiasis. Since in the retrospective study on our current clinical practice [19], we observed an effect of nutrition on stone formation, we then chose to restrict the variability of nutritional regimens, using low salt content, high vitamin C, and high protein. This clinical orientation may prevent the possibility of performing observational research on our nutrition regimens.

Given the above limitations, we believe it would be unadvisable to suggest clinical guidance based on dietary adjustments, in the absence of an experimental approach. A future study designed to provide clinical guidance should be larger, multicentric, interventional, and should include dietary comparison groups, or dietary intervention aside with pharmacological treatment, and check whether urolithiasis can be prospectively resolved by optimizing nutrition parameters. Moreover, a panel of exams including fluid balance and electrolyte excretion should be performed, not limited to those allowed in the clinical routine. Since different kinds of stones (oxalate, struvite) are created with different etiopathogenic mechanisms, studies should analyze with more specificity the causes of formation of different kinds of stones, separating cohorts of patients based on the stone type.

## 5. Conclusions

Clinical conclusions which can be derived from our work are that patients recovering from brain injury must be closely monitored for the possibility of urolithiasis if they have a lower GCS, if they need catheterization, if they practiced heavy or competitive sports, if they have higher blood potassium and urine pH values. Nutrition regimens rich in vitamin C and proteins, but not in electrolytes, seem to be neutral with respect to urolithiasis risk.

## Figures and Tables

**Table 1 nutrients-17-00883-t001:** Prevalence of urinary tract infections.

Microorganism	Whole Cohort	Patients with Urolithiasis
*E. faecalis*	6 (12.5%)	2 pre-existing
*E. coli*	4 (8.3%)	1 pre-existing 1 new occurrence
*K. pneumoniae*	4 (8.3%)	2 new occurrences
*P aeruginosa*	2 (4.2%)	0
*C. koseri*	1 (2.1%)	1 pre-existing
*M. morganii*	1 (2.1%)	0

**Table 2 nutrients-17-00883-t002:** Reference daily nutrition intakes in milligrams per day.

Patient Age	Na	K	Mg	Ca	Proteins	Vitamin C
6–12 mo	200	750	80	280	660/kg	20
1–3 y	1100	800	170	450		20
4–6 y	1300	1100	230	800		30
7–10 y	1700	1800	230	800		45
11–14 y	2000	2700	300	1150		70
15–17 y	2000	3500	300	1150		100

Sources for reference values: [22,23,24,25,26,27].

**Table 3 nutrients-17-00883-t003:** Comparison of clinical variables, nutrition variables, and blood and urine exam results between patients with or without urolithiasis.

Clinical Variable		No Urolithiasis		Urolithiasis		*p* ^1^
	Unit	Mean	SD	Mean	SD	
Age	years	8.4	1.1	12.6	1.5	*0.050*
Weight	kg	26.7	3.2	38.0	4.5	0.06
ABI etiology						0.18
Trauma		22.9%		46.2%		
Anoxia/ischemia		22.9%		23.1%		
Stroke		11.4%		23.1%		
Encephalopathy		31.4%		7.7%		
Other		11.4%		0%		
GCS—acute	\	5.7	3.3	6.2	3.8	0.67
GCS—subacute	\	10.0	0.4	9.3	0.8	0.43
Tracheostomy	yes	42.9%		76.9%		*0.036*
Oral feeding	yes	8.6%		0%		0.43
Catheterism	yes	14.3%		38.5%		0.19
Gallstones	yes	8.6%		15.4%		0.19
Intense physical activity before ABI	yes	28.6%		69.2%		*0.037*
Bedridden after ABI	yes	94.3%		100%		0.68
Nutrition						
Water intake	mL/d	614	277	908	444	*0.008*
Na	mg/kg/d	39.2	29.3	31.9	21.3	0.88
K	mg/kg/d	49.1	18.9	42.9	18.9	0.55
Mg	mg/kg/d	5.5	2.5	5.6	3.0	0.31
Ca	mg/kg/d	33.1	18.9	29.9	14.5	0.65
Vitamin C	mg/kg/d	5.0	2.9	3.8	1.5	0.14
Proteins	mg/kg/d	1.3	0.6	1.1	0.3	0.40
Blood exams						
Na	mmol/L	139.4	4.8	139.2	2.7	0.88
K	mmol/L	4.3	0.5	4.4	0.4	0.56
Cl	mmol/L	101.6	4.2	100.5	2.7	0.38
Ca	mmol/L	9.8	0.8	9.9	0.6	0.65
Mg	mmol/L	2.0	0.3	1.9	0.2	0.31
P	mmol/L	4.7	1.5	4.8	0.6	0.89
Parathormone	pg/mL	10.6	10.3	5.2	2.9	0.15
Vitamin D	ng/mL	20.5	10.1	43.0	57.5	0.22
Alkaline phosphatase	U/L	159.7	93.5	107.7	43.0	0.10
Uric acid	mg/dL	3.4	1.3	3.3	1.7	0.95
Urine exams						
Urine pH	\	7.0	0.7	7.5	0.5	*0.024*
Urinary tract infections	yes	31.4%		53.8%		0.15
Antiepileptic drugs use						
Lamotrigine daily dose	mg/kg	0	0	0.17	0.97	0.33
Levetiracetam daily dose	mg/kg	19.9	2.7	16.6	2.9	0.26
Oxcarbazepine daily dose	mg/kg	0.64	4.64	0.41	2.34	0.79
Phenobarbital daily dose	mg/kg	1.1	2.5	0.5	1.3	0.10
Topiramate daily dose	mg/kg	0.11	0.83	0.30	1.72	0.51
Valproic acid daily dose	mg/kg	4.0	12.0	1.3	7.2	0.18

^1^ *p*-values refer to results of *T*-tests or chi-squared tests. Values reported in italics are significant.

**Table 4 nutrients-17-00883-t004:** Retained variables associated with development of urolithiasis.

Variable	*p*-Value	OR ^1^	95% C.I.s
subacute GCS	0.007	0.68	0.51–0.90
Intense physical activity before ABI: yes	0.012	5.12	1.43–18.40
K blood levels mmol/l	0.021	4.41	1.26–15.48
Catheterism: yes	0.030	5.59	1.18–26.58
Urine pH	0.048	2.40	1.01–5.74

^1^ ORs are referred to changes of one point, regardless of the unit of measure.

## Data Availability

The dataset is available from the corresponding author upon request for non-profit purposes. The data are not publicly available due to privacy reasons.

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
