# Peer review of "Prospective Study of Urinary Stone Formation in Pediatric Patients with Acquired Brain Injury: A Focus on Incidence and Analysis of Risk Factors"

_nutrients, 2025, doi:10.3390/nu17050883_

Round 1

Reviewer 1 Report

Comments and Suggestions for Authors

I had the great privilege to review the manuscript entitled “Prospective study of urinary stone formation in pediatric patients with acquired brain injury, focus on incidence and analysis of risk factors” However, the manuscript can be improved by addressing the following points that the authors should consider:

  1. Although this study incorporated retrospective data, the prospective observational component lacked a control group that did not receive enteral nutrition. As a result, it is difficult to directly compare the risk of urolithiasis associated with enteral nutrition. Including a matched healthy control group or patients on different nutritional regimens would provide a clearer assessment of the impact of specific nutrients on stone formation.
  2. Since this study used retrospective data and an observational design, the findings only indicate associations between variables but do not establish causality. For instance, while a higher urine pH is linked to increased urolithiasis risk, it is unclear whether urine pH directly causes stone formation or if other underlying factors mediate this relationship. Future research should consider randomized controlled trials (RCTs) or propensity score matching (PSM) to strengthen causal inference.
  3. This study examined the effects of sodium, calcium, vitamin C, and protein on urolithiasis risk but did not clearly identify which nutrients were primary contributors or protective factors. Additionally, while the study found no significant association between sodium intake and stone formation, this may be due to changes in institutional nutrition protocols rather than sodium itself being irrelevant. Further analysis of urinary electrolyte excretion patterns (e.g., urinary calcium, sodium, and citrate levels) would help clarify which nutrients influence stone formation.
  4. The study used logistic regression to evaluate risk factors for urolithiasis, but its pseudo R² was only 0.47, indicating that more than 50% of the variability remains unexplained. Other unaccounted-for factors, such as hydration levels, fluid balance, and medication use (e.g., anticonvulsants), may influence stone formation. Further investigation into the interaction effects of these variables could improve the predictive accuracy of the model.
  5. This study found that higher urine pH was associated with urolithiasis, but the underlying mechanism was not fully explored. For example, calcium phosphate stones tend to form more readily in alkaline conditions, and elevated urine pH may be linked to altered renal excretion or dietary factors. Future research should consider measuring urinary citrate, oxalate, and calcium excretion to better understand the role of urine pH in stone formation.
  6. The study concludes that enteral nutrition rich in vitamin C and protein but low in electrolytes does not significantly impact urolithiasis risk, but it lacks specific clinical recommendations. For instance, should high-risk patients (e.g., those with a history of urolithiasis) receive targeted dietary modifications? Further discussion is needed on how these findings can be translated into improved nutritional management strategies for pediatric brain injury patients to minimize the risk of urolithiasis.

Author Response

Reviewer 1

I had the great privilege to review the manuscript entitled “Prospective study of urinary stone formation in pediatric patients with acquired brain injury, focus on incidence and analysis of risk factors” However, the manuscript can be improved by addressing the following points that the authors should consider:

  • Although this study incorporated retrospective data, the prospective observational component lacked a control group that did not receive enteral nutrition. As a result, it is difficult to directly compare the risk of urolithiasis associated with enteral nutrition. Including a matched healthy control group or patients on different nutritional regimens would provide a clearer assessment of the impact of specific nutrients on stone formation.

We thank the Reviewer for the methodological suggestion. We acknowledge in the discussion that this would improve the quality of the prospective study. Up to now, we did not follow patients without enteral nutrition since, in our previous work, we observed in the sizable retrospective cohort that in the absence of enteral nutrition the occurrence of urolithiasis was none. We therefore chose to analyze the components of nutrition mixtures.

  • Since this study used retrospective data and an observational design, the findings only indicate associations between variables but do not establish causality. For instance, while a higher urine pH is linked to increased urolithiasis risk, it is unclear whether urine pH directly causes stone formation or if other underlying factors mediate this relationship. Future research should consider randomized controlled trials (RCTs) or propensity score matching (PSM) to strengthen causal inference.

We agree with the Reviewer and mention this limitation in the discussion.

  • This study examined the effects of sodium, calcium, vitamin C, and protein on urolithiasis risk but did not clearly identify which nutrients were primary contributors or protective factors. Additionally, while the study found no significant association between sodium intake and stone formation, this may be due to changes in institutional nutrition protocols rather than sodium itself being irrelevant. Further analysis of urinary electrolyte excretion patterns (e.g., urinary calcium, sodium, and citrate levels) would help clarify which nutrients influence stone formation.

We thank the Reviewer for the suggestion and mention this limitation in the discussion. We will consider this improvement in future studies.

  • The study used logistic regression to evaluate risk factors for urolithiasis, but its pseudo R² was only 0.47, indicating that more than 50% of the variability remains unexplained. Other unaccounted-for factors, such as hydration levels, fluid balance, and medication use (e.g., anticonvulsants), may influence stone formation. Further investigation into the interaction effects of these variables could improve the predictive accuracy of the model.

We agree with the Reviewer and indicate in the discussion that the regression model is lacking. We have no detailed data on fluid balance. However, with respect to anticonvulsants, we have available data on drug use. Therefore, we reported data on the use of antiseizure drugs in table 3 and included in the regression model the daily prescribed dose of levetiracetam, lamotrigine, phenobarbital, phenytoin, oxcarbazepine, topiramate, valproic acid. Results indicate that none of these drugs is in relationship with the occurrence of urolithiasis in either correlation or regression statistics, therefore the final R2 did not increase. We added these insights in the results and discussion sections.

  • This study found that higher urine pH was associated with urolithiasis, but the underlying mechanism was not fully explored. For example, calcium phosphate stones tend to form more readily in alkaline conditions, and elevated urine pH may be linked to altered renal excretion or dietary factors. Future research should consider measuring urinary citrate, oxalate, and calcium excretion to better understand the role of urine pH in stone formation.

We agree with the Reviewer that data on urinary excretion are important; the present study had an observational nature and we could not prescribe exams for the purpose of research. Future studies should consider an interventional design. We have inserted these suggestions in the limitations section.

  • The study concludes that enteral nutrition rich in vitamin C and protein but low in electrolytes does not significantly impact urolithiasis risk, but it lacks specific clinical recommendations. For instance, should high-risk patients (e.g., those with a history of urolithiasis) receive targeted dietary modifications? Further discussion is needed on how these findings can be translated into improved nutritional management strategies for pediatric brain injury patients to minimize the risk of urolithiasis.

It is important to stress that in our previous study we observed only patients with enteral nutrition developing urolithiasis. No patient without enteral nutrition developed urolithiasis. This complete correlation indicates that either enteral nutrition is in itself a critical risk factor for urolithiasis, or that the time-frame of recovery during which enteral nutrition can be used, i.e. the sub-acute phase, is the critical one for the development of urolithiasis.

Since in the retrospective study on our current clinical practice, we observed an effect of nutrition on stone formation, we then chose to restrict the variability of nutritional regimens, using low salt content, high vitamin C and high protein. This clinical orientation may prevent the possibility to perform observational research on our nutrition regimens.

Given the above limitations, we believe it would be unadvisable to suggest clinical guidance based on dietary adjustments, in the absence of an experimental approach. A future study should include dietary comparison groups, or dietary intervention aside with pharmacological treatment and check whether urolithiasis can be prospectively resolved by optimizing nutrition parameters.

We reported this important issue in the limitations section.

Reviewer 2 Report

Comments and Suggestions for Authors

This prospective observational study investigates the incidence of urinary stone formation in pediatric patients with acquired brain injury and finds that enteral nutrition does not significantly contribute to stone formation. A few issues need to be addressed before acceptance can be considered.

1. The logistic regression model explains only 47% of the variability in urolithiasis occurrence. What other potential risk factors were not considered, and how could the model be improved to enhance predictive accuracy?

2. The study identifies a significant association between higher urine pH and stone formation but does not explore the underlying mechanisms. 

3. With a total sample of 88 patients (48 prospective and 40 retrospective), how representative is this study of the broader pediatric ABI population, and what steps could be taken to validate these findings in a larger, multicenter cohort?

Author Response

Reviewer 2

This prospective observational study investigates the incidence of urinary stone formation in pediatric patients with acquired brain injury and finds that enteral nutrition does not significantly contribute to stone formation. A few issues need to be addressed before acceptance can be considered.

  1. The logistic regression model explains only 47% of the variability in urolithiasis occurrence. What other potential risk factors were not considered, and how could the model be improved to enhance predictive accuracy?

We agree with the Reviewer that this is an important limitation to be considered and we discussed in the manuscript about the addition of other associated factors, regarding for instance electrolyte excretion, drug therapy and more.

  1. The study identifies a significant association between higher urine pH and stone formation but does not explore the underlying mechanisms. 

Higher urine pH is described in the literature to maintain the binding of calcium ions to phosphate ions, forming a stable calcium phosphate salt that can build up in stones. At acidic pH, the calcium phosphate salts are more likely to dissociate and do not build up. We inserted this mechanistic detail in the discussion.

  1. With a total sample of 88 patients (48 prospective and 40 retrospective), how representative is this study of the broader pediatric ABI population, and what steps could be taken to validate these findings in a larger, multicenter cohort?

We agree with the Reviewer that a larger and interventional study is required, including more exams than those allowed in the clinical routine. The study should be multicentric and involve nutritional intervention. We added this perspective in the discussion section.

Reviewer 3 Report

Comments and Suggestions for Authors

The authors of the article analyze the cases and risk factors of urinary stone formation in a strictly defined group of patients, i.e. pediatric patients with brain damage. The results of the analyses showed that 12-14% of patients had urolithiasis and the factors predisposing to their development did not differ much from those known in other research groups. Importantly, nutritional mixtures administered enterally were not shown to have an effect on the development of urolithiasis. The authors collected a lot of data and analyzed it properly. However, I have some comments and remarks which I will post below.

- whether the mineral composition of urinary stones in the patients studied is known? Information was given that the stones were primarily calcium and phosphate based. Was the presence of struvite detected and in what percentage? Urine pH and urinary tract infections could suggest this

- In the introduction, the authors mention the most common oxalate stones and the less common struvite stones. Each of them has a different basis of formation and there will be other risk factors too. Patients with struvite stones with infection may constitute a separate group here

- What treatment regimen was adopted in patients with urinary tract infections? In the undertaken analysis, urinary tract infections and catheterization were very important factors influencing the development of stones. Did the infection occur simultaneously with the appearance of urinary stones or was it secondary?

-species names of microorganisms should be written in italics

Author Response

Reviewer 3

The authors of the article analyze the cases and risk factors of urinary stone formation in a strictly defined group of patients, i.e. pediatric patients with brain damage. The results of the analyses showed that 12-14% of patients had urolithiasis and the factors predisposing to their development did not differ much from those known in other research groups. Importantly, nutritional mixtures administered enterally were not shown to have an effect on the development of urolithiasis. The authors collected a lot of data and analyzed it properly. However, I have some comments and remarks which I will post below.

- whether the mineral composition of urinary stones in the patients studied is known? Information was given that the stones were primarily calcium and phosphate based. Was the presence of struvite detected and in what percentage? Urine pH and urinary tract infections could suggest this

Among the 6 patients with analyzable expelled stones, 3 had calcium oxalate stones (of which, 1 with E. coli infection), 1 struvite (with E. faecalis infection), 2 struvite and oxalate (one with E. faecalis, one with K. pneumoniae infection). We added this information to the results section.

- In the introduction, the authors mention the most common oxalate stones and the less common struvite stones. Each of them has a different basis of formation and there will be other risk factors too. Patients with struvite stones with infection may constitute a separate group here

We agree with the Reviewer. Due to the limited sample size in the present work, it would be meaningless to analyze 3 patients with struvite stones separately. Future studies should analyze with more specificity the causes of formation of different kinds of stones. We reported this suggestion in the discussion.

- What treatment regimen was adopted in patients with urinary tract infections? In the undertaken analysis, urinary tract infections and catheterization were very important factors influencing the development of stones. Did the infection occur simultaneously with the appearance of urinary stones or was it secondary?

In view of the importance of infections, we inserted another table in the manuscript text, detailing further the infection status of patients in the whole cohort and in the group of patients with stones. Numbers are too limited to draw conclusions on temporal associations. With respect to treatment, only 2 patients were taking antibiotics at admittance and they did not develop stones.

It is important to highlight that in our clinical practice, potential cases of urinary tract infection are managed together with urologist consultancy and, in the absence of symptoms, we follow the principle of avoiding unnecessary antibiotic treatment. This approach has the advantage of reducing the spread of antibiotic resistance and reducing adverse effects.

Alternatives to systemic antibiotic therapy that we often consider are intravesical instillation of antibiotic, improved hydration, and change of urine pH to make the microenvironment unfavorable for bacteria.

Antibiotic treatment is started only in the presence of fever, blood markers of inflammation, positive abdominal echography, urine analysis with bacterial load above 100000 cfu/ml, and provided the antibiogram suggests viable strategies. The only exception is for emergency situations when rapid intervention with a broad spectrum antibiotic is advised.

We added these details to the discussion section in view of the importance of the suggested topic.

-species names of microorganisms should be written in italics

We thank the Reviewer for providing this correction.

Round 2

Reviewer 3 Report

Comments and Suggestions for Authors

I would like to thank the authors for responding to my comments and improving the manuscript in accordance with my remarks. I have no more comments on this article.